# Morpho-agronomic variability of okra [*Abelmoschus esculentus* (L.) Moench] genotypes in Dire Dawa, eastern Ethiopia

**Wubadis Kenaw** [ORCID]*, **Wassu Mohammed, Kebede Woldetsadik**

School of Plant Sciences, Haramaya University, Dire Dawa, Ethiopia

* wubadiskenaw2009@gmail.com, Wubadis.Kenaw@haramaya.edu.et

**Data Availability Statement:** All relevant data are within the manuscript.

## Abstract

A total of 21 okra genotypes were evaluated for 25 morpho-agronomic traits in 2020 at Dire Dawa, Ethiopia in a randomized complete block design with three replications. Analysis of variance showed significant differences at p<0.05 level of significance for all traits. Estimates of genotypic (GCV) and phenotypic (PCV) coefficients of variation range from 9.16 to 42.3% and 9.33 to 44.16%, respectively. Heritability in a broad sense ($H^2$) and genetic advance as a percent of the mean (GAM) ranged from 29.57 to 91.89% and 10.39 to 83.53%, respectively. Estimated variability components (GCV, PCV, $H^2$, and GAM) were high and moderate for all traits except days to 50% emergence 9.33% of GCV and PCV, internode length 9.16% of GCV and green fruit width 29.57% of $H^2$ that were categorized under low. The first four principal component axes (PCA1 to PCA4) accounted for 7.83 to 35.02%, which accounted 74.56% of the total variability with eigenvalues that ranged from 1.95 to 8.75. Genetic distances estimated by Euclidean distance from the 25 traits ranged from 2.33 to 12.56 with a mean of 6.83, standard deviation of 1.8, and a coefficient of variation of 26.46%. The genotypes were grouped into four distinct clusters using the Euclidean distance matrix using UPGMA. Indigenous okra genotypes collected from Ethiopia were more divergent with high genetic distances and had a higher performance for most of the traits including growth, green fruit yield, and seed yield than introduced genotypes. In conclusion, this study showed the presence of variation among genotypes for most of the traits, indicating that selection of genotypes could be effective to develop okra varieties with high green fruit and seed yield through direct selection or crossing.

## Introduction

Okra [*Abelmoschus esculentus* (L.) Moench] is one of the most widely utilized species of the family *Malvaceae* [1]. It originated in Ethiopia and is native to the areas of Ethiopia and Sudan [2]. It is cultivated as a vegetable crop in tropical, subtropical, warm, and temperate regions of the world [3]. It is an important crop in most developing countries, covering approximately 4% of the total vegetable consumption, and it ranks above other vegetable crops including cabbage, amaranths, and lettuce [4,5].

Okra is mainly grown for its immature pods, which are consumed as a fresh vegetable, a snack, a cooked vegetable, or as an additive for soups, salads, and stews. It is a good source of

**Funding:** The author(s) received no specific funding for this work.

**Competing interests:** The authors have declared that no competing interests exist.

carbohydrates, protein, dietary fibre, calcium, magnesium, potassium, vitamins A and C, fat, iron, thiamine, nicotinamide, and riboflavin [6]. Okra seed is an excellent source of zinc, and rich in phenolic compounds, has a balanced protein of both lysine and tryptophan amino acids which is comparable to soybean [7]. Seed flour is used to fortify cereal flour to increase the meal's protein, ash, oil, and fibre content. The potential for wide cultivation of edible oil as well as for cakes is very high—because the oil content of the seed is high at about 40% [8] and the oil is a rich source of linoleic acid, a polyunsaturated fatty acid essential for humans nutrition, rich in palmitic acid and oleic acids [9]. The dried seeds can be used to prepare vegetable curds and as a substitute and/or coffee additive after roasted. These properties enhance the importance of okra, pod, and seed foodstuff in the human diet [10].

Okra is a perfect candidate for sustainable agriculture; the crop should be attractive to both producers and consumers in terms of its growth, yield, and quality. Considerably high magnitude of genetic variability of vigorous growth, earliness, and yield-associated traits have been reported in okra germplasm [11]. This high genetic variability is a key to okra improvement. The existing variability has been exploited in various breeding programs, which have resulted in the development and release of a good number of varieties of okra. However, the released varieties cannot be continued longer due to the development and replacement of traditional crops by new high -yielding varieties and the impact of climate change.

The first step for the improvement of indigenous vegetable crops is to study the genetic variability in the available genotypes. This is because the high genetic variability of genotypes gives a better opportunity for breeders to select directly for the traits of interest [12]. The genetic variability of okra landraces has been studied in Ethiopia [13–17]. The information generated from past studies gave a clue to the potential of improving okra genotypes. But the studies were focusing on assessing the genetic diversity of okra genotypes for varied traits with respect to their diverse geographical origins. However, information on genetic diversity alone is not sufficient for the improvement of okra landraces for yield and fruit quality. The improvement of the crop requires continuous evaluation of genotypes for fruit yield, quality, and other morpho-agronomic traits, comparing the potential of indigenous with introduced varieties to identify and release varieties with a short period as possible; and evaluating available okra indigenous and introduced varieties for stress tolerance to ascertain the crop's resilience to climate change. The recently conducted research tried to generate information about the genetic variability among okra genotypes only in the major okra growing regions in Ethiopia. But it is necessary to generate genetic variability of okra genotypes and morphological differences among okra genotypes from both introduced and local collections that can be exploited in breeding programs to develop high -yielding varieties and improve their quality [18].

Therefore, it was necessary to continue the evaluation of available indigenous and introduced okra varieties for yield, fruit quality, and its components to estimate the genotype potential to withstand stresses such as high temperatures and to identify superior okra genotypes to be used in breeding programs. Thus, this study was conducted to assess morpho-agronomic traits for fruit yield variations among indigenous and introduced okra genotypes and the genetic variability of indigenous and introduced okra genotypes.

## Materials and methods

### Description of the study site

The experiment was conducted at Tony farm, the research site of Haramaya University at Dire Dawa, from February to June 2020 under irrigation conditions. Dire Dawa lies between latitudes 9˚36′N, 41˚52′E and characterized by a warm and dry climate with a relatively low level

of precipitation. It has an altitude of 1260 meters above sea level, a mean annual temperature of about 24.8˚C, an average maximum temperature of 31.4˚C, and an average minimum temperature of about 18.2˚C. The aggregate average annual rainfall is about 604mm, and the annual relative humidity is 41.82% [19].

## Experimental materials and design

A total of 21 okra genotypes, 11 okra genotypes collected from four geographic regions (western, south-western, north-western, and northern Ethiopia) known as the major growing regions of okra in the country, and 10 commercial varieties introduced from India and USA were evaluated in the present study. The four major growing regions of okra are represented by 5 genotypes (240586, 240609, 240591, 245157, and 242443) and 4 (240209, 92203, 240207, and 242433) genotypes collected from 490 to 935 and 1050 to 1480 meters above sea level, respectively. The nine genotypes were collected from nine known districts, such as Akobo (490m), Gambella, Akobo (630m), SirbAbaye, Menge, Mandura, DigaLeka, Dibate, and Asossa, and two genotypes Humera 01 and (*Bamaya-Humera*) collected from northern Ethiopia. Two (SOH 714 and SOH 701) commercial varieties introduced and registered by companies, seven commercial varieties introduced from India, and one commercial variety introduced from the USA for research purposes were used for the study "Table 1".

The field experiment was conducted in a randomized complete block design with threereplications. Each genotype was randomly assigned to a plot in each replication in the experimental field. Each plot consisted of 12 plants at a spacing of 0.6 m between plants. The spacing

**Table 1. List of 21 Okra genotypes and their origin.**

| No | Accession code | Regional state | District | Altitude m.a.s.l. | Geographic region |
|----|----------------|----------------|----------|-------------------|-------------------|
| 1 | 240586 | Gambella | Akobo | 490 | Southwestern |
| 2 | 240609 | Gambella | Gambella | 730 | Southwestern |
| 3 | 240591 | Gambella | Akobo | 630 | Southwestern |
| 4 | 240209 | BenishangulGumuz | Mandura | 1050 | North-western |
| 5 | 240207 | BenishangulGumuz | Dibate | 1400 | North-western |
| 6 | 92203 | Oromia | DigaLeka | 1200 | Western |
| 7 | 242433 | BenishangulGumuz | Asossa | 1480 | Western |
| 8 | 245157 | BenishangulGumuz | SirbAbaye | 870 | Western |
| 9 | 242443 | BenishangulGumuz | Menge | 935 | Western |
| 10 | Humera 1 | Tigray | Northern | | |
| 11 | 23793 (Bamaya-Humera) | Tigray | Released variety | | Northern |
| 12 | SOH 714 | | Registered commercial variety | | |
| 13 | SOH 701 | | Registered commercial variety | | |
| 14 | Dhenu | | India | | |
| 15 | Kiran | | India | | |
| 16 | Kraft | | | | India |
| 17 | Pocha | | | | India |
| 18 | Mithra | | | | India |
| 19 | Arka Anamica | | | | India |
| 20 | NamdHari | | | | India |
| 21 | Clemson Spineless | | | | USA |

Source of genotypes: Haramaya University and Institute of Biodiversity and Conservation Research, Addis Ababa.

between the plots and adjacent replications was 0.8 and 2 m, respectively. Three seeds per hole were sown and thinned to one plant per hole when plants reached 3–4 leaf stages.

## Experimental procedures and plot management

The seeds were harvested in the January and February 2019 cropping seasons at Mellkasa Agriculture Research Centre, and kept in canvas bags after drying and stored at the Institute of Biodiversity and Conservation Research, Addis Ababa.

Land preparation was done in the first week of February 2020 using a tractor and human labor. The soil was levelled to permit furrow irrigation every five days. The rows were raised to increase soil surface area, aeration, and drainage. The ridges were made according to the plant spacing's by hand. Okra seeds were sown in mid-February 2020 and placed at a depth of 5 cm. Furrow irrigation was applied every five days. Cultural practices such as cultivation, hoeing, weeding, and earthing-up were applied uniformly.

## Data collection

Data were collected for phenology traits (days to 50% emergence, days to first flowering, days to 50% flowering, and days to maturity), growth, and yield- related traits (plant height, stem diameter, number of branches per plant, number of internodes, internodes length, leaf length, leaf width, number of green fruits, the weight of green fruits per plant, green fruit length, green fruit width, average green fruits weight, fruit yield per hectare, number of dry pods, the weight of dry pods per plant, average dry pod weight, number of seeds per pod, hundred seeds weight, moisture content of seeds, seed weight per pod, and seed yield per plant. Data on quantitative traits were recorded according to the International Plant Genetic Resources Institute (IPGRI, 1991) descriptor list developed for okra.

## Data analysis

The quantitative data were subjected to analysis of variance (ANOVA) and computed with SAS statistical software (version 9.0). The data recorded as percentages were transformed before subjecting them to the analysis of variance [20]. The comparison of the mean performance of genotypes was conducted following the significance of mean squares using the least significant difference at 5% probability level. The traits that exhibited significant mean squares in general ANOVA were further subjected to multivariate analysis.

## Principal component analysis

The morpho-agronomic traits that exhibited significant mean squares in general ANOVA were pre-standardized to means of zero and variance of unity before principal component analysis (PCA) was performed to avoid bias due to differences in measurement scales [21]. After standardization, the data were subjected to principal component (PC) analysis to understand which trait(s) most contributed to the divergence or total variability of genotypes.

## Genetic distance and clustering

The morpho-agronomic traits that exhibited significant mean squares in general ANOVA were used to estimate genetic distances and clustering of genotypes. Genetic distances of 21 okra genotypes were estimated using Euclidean distance (ED) calculated from quantitative traits after standardization (subtracting the mean value and dividing it by the standard deviation) as established by Sneath [5].

**Table 2. Mean squares from the analysis of variance for crop phenology and growth traits of 21 okra genotypes evaluated at Dire Dawa in 2020.**

| Trait | Replication (2) | Genotype (20) | Error (40) | CV (%) |
|---|---|---|---|---|
| Days to 50% emergence | 2.90 | 4.87** | 1.73 | 12.04 |
| Days to first flowering | 9.82 | 226.11** | 82.34 | 17.02 |
| Days to 50% of flowering | 35.00 | 359.90** | 120.20 | 16.90 |
| Days to maturity | 168.06 | 667.37*** | 164.44 | 16.12 |
| Plant height (cm) | 190.90 | 1335.49*** | 106.39 | 6.68 |
| Stem diameter (cm) | 0.04942 | 0.56708*** | 0.12532 | 10.50 |
| Number of branches per plant | 0.07476 | 5.08371*** | 0.37426 | 15.20 |
| Number of internodes | 4.2433 | 46.7894*** | 8.7483 | 9.42 |
| Internode length (cm) | 0.60619 | 0.80786*** | 0.19069 | 8.83 |
| Leaf length (cm) | 54.8523 | 17.4999*** | 3.9692 | 10.29 |
| Leaf width (cm) | 84.2716 | 41.0830** | 13.4276 | 15.39 |

**and *** significant at P<0.01 respectively. CV (%) = Coefficient of variation in percent. Numbers in parenthesis represent degree of freedom for the respective source of variation.

## Results and discussion

### Analysis of variance

The results of the analysis of the variance for phenology, growth, yield-related traits, dry pod, and seed-related yield traits showed a significant (P < 0.05) difference of 21 okra genotypes are presented in "Tables 2 and 3".

The results indicated that the presence of significant variations among genotypes for quantitative traits might provide a good opportunity for breeders to select genotypes with varied crop maturity and better growth performance, high-yielding and most preferable genotypes,

**Table 3. Mean squares from the analysis of variance for yield-related traits, dry pod, and seed-related yield traits of 21 okra genotypes evaluated at Dire Dawa in 2020.**

| Trait | Replication (2) | Genotype (20) | Error (40) | CV (%) |
|---|---|---|---|---|
| Number of green fruits per plant | 1.4885 | 56.7957*** | 1.9923 | 9.33 |
| Weight of green fruits per plant (g) | 1785.8 | 47672.6*** | 1834.4 | 14.30 |
| Green fruit length (cm) | 10.0340 | 50.4439*** | 5.5326 | 15.95 |
| Green fruit width (cm) | 0.04886 | 0.20984* | 0.09295 | 14.33 |
| Average green fruits weight (g) | 41.446 | 263.243*** | 24.758 | 13.98 |
| Fruit yield per hectare (t ha$^{-1}$) | 0.7751 | 20.6906*** | 0.7961 | 14.30 |
| Number of dry pods per plant | 3.0895 | 56.7934*** | 5.6664 | 15.14 |
| Weight of dry pod per plant(g) | 24779 | 164032*** | 8100 | 12.92 |
| Average dry pod weight (g) | 85.84 | 1208.42*** | 34.53 | 12.57 |
| Number of seeds per pod | 131.16 | 1339.02*** | 114.20 | 13.51 |
| Hundred seeds weight (g) | 0.18476 | 3.26967*** | 0.32643 | 9.42 |
| Seed moisture content (%) | 5.87648 | 9.98627*** | 1.99700 | 13.98 |
| Seeds weight per pod (mg) | 404838 | 6773952*** | 458105 | 13.21 |
| Seed yield per plant (g) | 6.91 | 1117.19*** | 34.51 | 13.05 |

*and *** Significant at P<0.05 and P<0.01, respectively. CV (%) = Coefficient of Variation in percent. Numbers in parenthesis represent degree of freedom for the respective source of variation.

and hybridization or selection of traits of interest. In his review, Alemu [18] found that for the quantitative features examined, there were highly significant variances across the 35 indigenous okra collections. He also mentioned that there was variation in the trait collections among the Ethiopian okra landraces. There were highly significant differences among the 63 okra genotypes for all examined agronomic (growth), phenological, yield, and yield-related traits [22]. Mohammed et al. [15] reported that there were significant differences in the number of fruits per plant, green fruit weight, fruit length, and fruit width. According to Muluken et al. [13,15–17] also reported similar significant differences for the number of pods per plant, the weight of dry pods per plant, the average weight of dry pods, number of seeds per pod, seed weight per pod, hundred seed weight, and seed yield per plant among the genotypes evaluated.

## Mean performance of okra genotypes

The estimated wide range of variation in phenology and growth traits studied was observed in "Table 4". The mean values of genotypes were in the range of 9.33 to 12 days to 50% emergence, 46.99 to 60 days to first flowering, 55.35 to 75.91, and 66.64 to 96.5 days to 50% flowering and maturity, respectively.

The recently released variety 23793 (*Bamaya Humera*) and other genotype collected from northern Ethiopia (Humera 01) were earlier to attain 50% emergence, [1] days to flowering, and 50% flowering. Meanwhile, genotypes collected from western Ethiopia (242443, 92203, 245157, and 242433) have lately attained 50% emergence, [1] and 50% flowering, and days to maturity. The genotypes showed a wide range of variations in phonological traits. Okra genotypes were introduced from India were earlier in first flowering, 50% flowering and days to maturity than the indigenous varieties except northern collections. This earliness might be an advantage in areas where the rainy season is short, and the producer might help in capturing the early market, which fetches a high price in markets, and earn a high income by supplying fruit early where fruits are not harvested from other genotypes under irrigated production. The research results suggested a higher chance of developing okra varieties for better growth, green fruit yield, and other desirable traits either through selection, breeding, and/or crossing of genotypes.

In their review, Temam et al. [17] reported that significant variations for days to 50% emergence (7.5 to 11 days), days to first flowering (44.5 to 71 days), days to 50% flowering (48.5 to 77.5 days), and days to maturity (75.5 to 104.5 days). They also reported genotypes collected from western Ethiopia (Benishangul Gumuz) had mean performance greater than the mean

**Table 4. Mean values of 21 okra genotypes with respect to the geographic region for phenology and growth traits evaluated at Dire Dawa in 2020.**

| Geographic region | Emergence (50%) | 1st flowering | 50% flowering | Days to maturity | Plant height | Stem diameter | Number of branches | Leaf length | Leaf width | Number of internodes | Internode length |
|---|---|---|---|---|---|---|---|---|---|---|---|
| India | 11.33[a] | 46.99[b] | 55.47[b] | 66.64[c] | 162.26[ab] | 3.03[bc] | 2.93[b] | 17.38[b] | 21.16[bc] | 33.78[ab] | 4.80[b] |
| Northern | 9.33[c] | 46.50[b] | 55.35[b] | 67.00[c] | 137.02[cd] | 2.80[c] | 2.73[b] | 16.515[b] | 20.11[c] | 28.56[c] | 4.79[b] |
| North-western | 11.67[a] | 58.00[a] | 74.50[a] | 92.33[ab] | 148.94[bcd] | 3.51[a] | 5.05[a] | 19.61[ab] | 24.75[ab] | 27.995[c] | 5.35[a] |
| RCVs | 9.67[bc] | 54.65[ab] | 69.85[a] | 83.85[b] | 173.30[a] | 3.61[a] | 4.53[a] | 18.95[ab] | 22.56[bc] | 35.16[a] | 4.88[b] |
| South-western | 11.11[ab] | 60.00[a] | 69.57[a] | 84.47[b] | 130.93[d] | 3.77[a] | 4.57[a] | 21.34[a] | 27.79[a] | 27.28[c] | 4.95[a] |
| USA | 12.00[a] | 53.30[ab] | 62.3[ab] | 78.3[bc] | 180.08[a] | 3.39[ab] | 4.30[a] | 22.16[a] | 25.73[ab] | 31.80[abc] | 5.66[a] |
| Western | 11.00[ab] | 59.75[a] | 75.91[a] | 96.50[a] | 154.2[bc] | 3.74[a] | 5.31[a] | 21.37[a] | 26.94[a] | 32.30[bc] | 4.91[a] |

RCVs = Registered commercial variety, USA = United States of America, Alphabets = mean values within column with similar letter(s) had not significant differences.

values of the three tested genotypes for days to emergence. Mihretu et al. [4] also observed 37 to 65 and 50 to 101 days of first flowering and maturity, respectively, in 25 okra genotypes collected from south-western Ethiopia. This indicates early maturing okra genotypes can be useful to escape drought conditions and can be cultivated as a climate change crops in droughtprone areas in Ethiopia. According to the findings of YIMAM [16] variations for days to 50% flowering and days to maturity in the range of 64.18 to 89.84 and 73.33 to 105.67, respectively. The Okra plant usually sets its first flower one to two months after sowing and has a maturity duration of 90 to100 days. introduced okra genotypes had the lowest days to 1st and 50% flowering and days to maturity in their finding's indicating the introduced genotypes were early compared to okra genotypes collected from Ethiopia [13,23].

The mean values of genotypes introduced from India and the two registered commercial varieties and genotype introduced from the USA showed the highest plant heights of 162.26, 173.30, and 180.08cm, respectively. The two registered commercial varieties and genotype introduced from the USA also showed the highest stem diameter, number of branches per plant, leaf length, and number of internodes. While, the mean value of genotypes collected from northern Ethiopia showed the lowest values for all studied growth traits (plant height, stem diameter, number of branches, leaf length, leaf width, number of internodes, and internode length).

The result indicates that genotypes collected from southwestern, northwestern, northern, and western Ethiopia had a wider range of variations in growth traits except plant height and number of internodes than the introduced genotypes. Genotypes with long and wide leaves, a large number of internodes, and long internode length might contribute to higher photosynthesis activity, increase plant height, and thereby produce a higher fruit yield. Genotypes that have vigorous growth traits are mostly preferable to local farmers. In another way, tall plants lodge easily due to environmental factors such as excessive flooding, rain, and wind when they produce tall fruit during favorable growing seasons. However, they are also essential for firewood, the construction of houses and fences, and livestock feed. Furthermore, selection for the tall height genes might be important when yield performance is low under unfavorable environmental conditions. The genotype that had a tall gene had the highest mean values in the number of internodes, internode length, and thick stem. The thicker stem resists environmental influences from lodging and can withstand high fruit yields [24].

In her findings, YIMAM [16] observed significant variations among okra genotypes for plant height (175.2 cm), stem diameter (2.67cm), and internode length (8.99cm) traits. She also reported that okra genotypes collected from Ethiopia showed more variation in growth traits compared to genotypes introduced from other countries. significant variations among okra genotypes for plant height and stem diameter in the range between 42.96 to 96 cm and 2.2 to 3.9 cm, respectively [4]. According to Muluken et al. [13] also reported 110.5 to 302.13 cm and 1.97 to 3.72 cm of plant height and stem diameter, respectively. They also reported okra genotypes collected from Ethiopia had higher mean values than the two exotic okra varieties for stem diameter and a number of branches.

The Ethiopian collections were superior to introduced genotypes for stem diameter, number of branches per plant, leaf length, and leaf width [23].There was a wide range of variations in growth traits of okra genotypes collected from Ethiopia [14]. Temam et al. [17] also reported variation among okra genotypes in a plant height that ranged between 73.05 to 194.5cm, stem diameter from 14.5 to 30.5mm, number of branches from 1.3 to 12.7, number of internodes from 13.5 to 39 and internode length from 2.5 to 10.25cm.

The estimated wide range of variation in fruit yield-related studied traits was presented in "Table 5". Genotypes collected from northern, north-western, and south-western regions and genotype introduced from UAS showed the highest mean values of the number of green fruits

**Table 5. Mean values of 21 okra genotypes with respect to the geographic region for green fruit yield and yield components evaluated at Dire Dawa in 2020.**

| Geographic region | No Green fruit per plant | Weight of Green Fruit per Plant (g) | Green Fruit Length (cm) | Green Fruit Width (cm) | Average Green Fruit weight(g) | Fruit yield (tons ha$^{-1}$) |
|---|---|---|---|---|---|---|
| India | 13.33[c] | 264.16[bc] | 14.84[ab] | 2.13[ab] | 33.69[c] | 5.50[bc] |
| Northern | 18.23[ab] | 414.44[a] | 13.27[ab] | 2.34[a] | 45.81[ab] | 8.63[a] |
| North-western | 19.81[a] | 361.80[ab] | 16.09[ab] | 2.19[ab] | 36.32[bc] | 7.53[ab] |
| RCVs | 9.36[d] | 174.20[c] | 16.67[a] | 2.14[ab] | 28.11[c] | 3.62[c] |
| South-western | 16.53[ab] | 336.41[ab] | 16.15[ab] | 2.13[ab] | 35.82[c] | 7.00[ab] |
| USA | 19.23[ab] | 475.45[a] | 10.21[b] | 2.44[a] | 49.47[a] | 9.90[a] |
| Western | 15.13[bc] | 263.34[bc] | 13.74[ab] | 1.86[b] | 33.48[c] | 5.48[bc] |

RCVs = Registered commercial variety, USA = United States of America. Alphabets = Mean values within column with similar letter(s) had not significant differences.

per plant, the weight of green fruits per plant, green fruit length, green fruit width, and fruit yield over the rest of the other genotypes. The research results showed the presence of a higher chance for selection of genotypes for longer and wider green fruits with higher green fruit weight, while the two registered commercial varieties, western Ethiopia and introduced from India, showed the lowest mean fruit yields of 3.62, 5.48, and 5.5 tons ha1, respectively. The genotypes had a wide range of green fruit yields. This result may be due to the cumulative effect of the number of green fruits per plant, green fruit weight, average fruit weight, and green fruit length, which show the genetic response of genotypes to environmental conditions. The longer and wider green fruits with a higher green fruit weight are preferred by consumers [25].

The current study result indicates that there is a wide variation in green fruit yield and yield component traits of okra genotypes collected from different parts of Ethiopia and introduced to other countries, which is a key solution to the current climate change concern [14]. Most of the genotypes collected from south-western, north-western, and northern Ethiopia had higher mean performance of green fruits yield and yield components than the introduced (India) and two registered commercial varieties. This showed that there is wide variability in green fruit yield and yield components in Ethiopia through the collection and selection of okra genotypes with the desired traits [23].

In their review, Muluken et al. [13] observed 25 okra genotypes and reported mean values of genotypes in the range of 25.65 to 71.67g, 6.95 to 20.73cm and 17.5 to 48.47 tons ha$^{-1}$ for fruit weight, fruit length and fruit yield with a mean value of 50.02g, 14.86 cm and 30.38 tons ha$^{-1}$, respectively. They also reported Ethiopian okra genotypes had higher mean values than the two exotic okra genotypes for number of fruit per plant and fruit yield per hectare. According to Binalfew et al. [14] means of genotypes ranging from 6.47 to 61.67g for average fruit weight indicating the variability among okra collections from Ethiopia. Temam et al. [17] also tested 36 okra genotypes and reported the average fruit weight and fruit yield per hectare of genotypes varied from 17.04 to 67.23g and 9.44 to 32.88tons, respectively.

The mean values of seed yield and related studied traits presented in "Table 6". Genotypes collected from north western (except hundred seed weight), western (except seed yield per plant) southwestern and two registered commercial varieties (except average dry pod and hundred seed weight) showed highest seed yield and related traits. The current result indicate that the highest number of seeds and hundred seed weight were considered as selection criteria for the breeding of okra genotypes for seed yield and yield component traits [24].

Genotypes collected from northern Ethiopia had the lowest mean values of seed moisture content (7.17%), while the remaining group of genotypes collected from northwestern,

**Table 6. Mean values of 21 okra genotypes with respect to the geographic region for seed yield and related traits evaluated at Dire Dawa in 2020.**

| Geographic region | No dry pod per plant | Weight of dry pod per Plant(g) | Average dry pod weight (g) | Number of seeds per pod | 100 seeds weight (g) | Seed moisture content (%) | Seeds weight per pod (mg) | Seed yield per plant (g) |
|---|---|---|---|---|---|---|---|---|
| India | 15.89[a] | 534.57[c] | 34.11[c] | 69.88[b] | 6.26[b] | 10.34[a] | 4580.47[b] | 41.51[bc] |
| Northern | 14.66[a] | 594.65[bc] | 41.73[bc] | 33.33[c] | 3.46[c] | 7.17[b] | 2032.50[c] | 14.72[d] |
| North-western | 17.66[a] | 897.57[a] | 51.06[ab] | 96.30[a] | 6.24[b] | 10.20[a] | 6045.90[a] | 61.98[ab] |
| RCVs | 17.68[a] | 721.00[abc] | 42.73[bc] | 88.16[a] | 6.06[b] | 11.27[a] | 5386.90[ab] | 61.30[a] |
| South-western | 17.05[a] | 895.43[a] | 53.06[ab] | 93.57[a] | 6.91[a] | 9.40[a] | 6483.56[a] | 60.66[a] |
| USA | 15.00[a] | 683.80[abc] | 45.70[bc] | 88.53[a] | 5.80[b] | 11.42[a] | 5168.90[ab] | 40.38[bc] |
| Western | 13.16[a] | 771.26[ab] | 66.78[a] | 91.88[a] | 6.33[ab] | 10.74[a] | 6005.30[a] | 39.09[c] |

RCVs = Registered commercial variety, USA = United States of America, Alphabets = mean values within column with similar letter(s) had not significant differences.

western Ethiopia, two registered commercial varieties and introduced from USA had higher seed moisture content from 9.4 to 11.42%. According to Bereded [26], the genotypes Clemson Spineless (introduced from USA) and ArkaAnamika (introduced from India) had the greatest seed moisture contents, measured 11.97 and 11.55%, respectively. Low moisture content in the seed is important to extend the shelf-life of okra seeds in storage place since selecting genotypes that had low seed moisture content is one of the smart post-harvest handling technologies to save farmers' economic losses. Genotypes collected from south-western Ethiopia (60.66g), the two registered commercial verities (61.30g), and genotypes collected from northern-western (61.98g) had higher mean values of seed yield per plant, while genotypes collected from northern, western Ethiopia, introduced from USA and India showed lower mean values of seed yield per plant 14.72, 39.09, 40.38 and 41.51g, respectively.

The observed high number of dry pods, number of seeds per pod, and hundred seed weight in okra genotypes suggested a higher chance of improving the seed yield of okra to produce a high amount of edible oil per unit area. The oil content of the seed is quite high at about 40% and the seeds are also used as a substitute for coffee [8].

In her review, YIMAM [16] reported genotypes had significantly different mean values for number of dry pods, weight of dry pods per plant, average dry pod weight, number of seeds per pod, seed weight per pod, hundred seed weight and seed yield per plant that ranged from 3.53 to 24.73, 175.9 to 1518.9 g, 12.55 to 83.93g, 110.17, 7.09g, 7.01g, and 5.86 to 153.89g respectively. The group of genotypes collected from Ethiopia had higher mean values than introduced genotypes for the number of mature pods per plant, weight of mature pods and hundred seed weight [15]. Genotypes had a wide range of variation 58.5 to 111.9, 5.08 to 7.16g, and 33.48 to 198.04g with mean values of 79.56, 5.93g and 99.7g for number of seeds per plant, hundred seed weight and seed yield per plant, respectively [17]. Okra genotypes collected from Ethiopia had the highest mean values for the number of seeds per pod and hundred seed weight [13]. Genotypes collected from different growing parts of Ethiopia are promising to be developed to variety with a high seed yield than introduced [4,13,14]. This suggests the a higher chance to improving seeds yield through selection among okra genotypes collected from Ethiopia to produce high amount of edible oil and for other purposes such as seeds as a substitute for coffee, protein source etc.

## Principal component analysis

The principal component analysis of 25 quantitative traits is presented in "Table 7". This principal component analysis resulted in four principal components (PCA1 to PCA4) with eigenvalues ranging from 1.95 to 8.75. The first four principal components accounted for a varied

**Table 7. The first four principal component axes for 25 quantitative traits of 21 okra genotypes were evaluated at Dire Dawa in 2020.**

| Traits | PCA1 | PCA2 | PCA3 | PAC4 |
|---|---|---|---|---|
| Days to 50% emergence | 0.264 | 0.271 | -0.022 | 0.514 |
| Days to first flowering | **0.836** | -0.221 | -0.372 | -0.020 |
| Days to 50% flowering | **0.917** | -0.093 | -0.224 | -0.091 |
| Days to maturity | **0.873** | -0.029 | -0.243 | -0.043 |
| Plant height (cm) | -0.040 | -0.198 | 0.445 | 0.572 |
| Stem diameter (cm) | **0.840** | 0.217 | 0.011 | -0.059 |
| Number of branches | 0.715 | 0.095 | 0.065 | -0.258 |
| Leaf length (cm) | **0.842** | 0.363 | 0.024 | 0.032 |
| Leaf width (cm) | **0.876** | 0.288 | -0.081 | 0.016 |
| Number of internodes | -0.125 | -0.507 | 0.488 | 0.007 |
| Internode length (cm) | 0.177 | 0.362 | -0.145 | **0.700** |
| Number of green fruits per plant | 0.072 | **0.904** | -0.038 | 0.036 |
| Weight of green fruit per plant (g) | -0.218 | **0.891** | 0.139 | -0.057 |
| Green fruit length (cm) | 0.048 | -0.611 | 0.033 | 0.132 |
| Green fruit width (cm) | -0.471 | 0.694 | 0.265 | 0.132 |
| Average green fruit weight (g) | -0.318 | **0.885** | -0.052 | -0.041 |
| Fruit yield per hectare (t ha-1) | -0.193 | **0.937** | 0.041 | 0.074 |
| Number of dry pods per plant | 0.056 | 0.136 | **0.803** | -0.404 |
| Weight of dry pods per plant (g) | 0.717 | 0.285 | 0.222 | -0.444 |
| Average dry pod weight (g) | 0.620 | 0.063 | -0.544 | -0.098 |
| Number of seeds per pod | **0.849** | 0.054 | 0.199 | 0.179 |
| Hundred seed weight (g) | 0.638 | -0.136 | 0.360 | 0.344 |
| Seed moisture content (%) | 0.447 | -0.202 | 0.518 | 0.357 |
| Seed weight per pod (mg) | **0.855** | 0.096 | 0.125 | 0.170 |
| Seed yield per plant (g) | 0.515 | 0.052 | 0.732 | -0.269 |
| Eigenvalue | 8.755 | 5.134 | 2.792 | 1.959 |
| Contribution (%) | 35.022 | 20.535 | 11.167 | 7.838 |
| Cumulative Variance (%) | 35.022 | 55.557 | 66.724 | 74.562 |

PCA1 to PCA4 represented the first principal component axis to the four principal component axes.

percentage of total variance that ranged from 7.83 to 35.02%, which summed to a total of 74.56% variation. The first four PCAs were retained in the analysis because each PCA had an eigenvalue >1. The PCAs with an eigenvalue of < 1.00 were ignored due to Gutten's lower bound principle that eigenvalues <1.00 should be ignored [27]. According to Demelie et al. [28] okra genotypes and reported PCAs with a cumulative contribution in the range of 64.32% to 83%.

The first four PCAs contributed to a total of 74.56% variance. The traits which had more contribution to total genetic divergence are under the control of additive gene action and will offer a good scope of improvement through selection breeding. The current study result indicates that direct selection can be practical for achieving desirable results. In their finding, Mohammed et al. [29] found the first principal component (PC1) accounted for the majority of the variation (30.54%), followed by PC2, PC3, and PC4, which each contributed 14.11%, 10.87%, and 6.98%, accounting for 62.51% of the overall variation. On the contrary, Demelie et al. [28] found the first three principal components (PCA1, PCA2, and PCA3) with values of 32.4%, 16.7%, and 8.2%, respectively, contributed more to the total of 57.3% variation.

Amoatey et al. [30] reported the first, second, and third principal components with values of 32.44%, 19.78%, and 9.68% of the total variance, respectively. The first principal component (PCA1) was 32.44% variance [31]. The output of PCA revealed that different traits contributed differently to the variation. These differences indicated the presence of variability and considerable opportunity for improvement in different quantitative traits. Principal component analysis (PCA) proved to be a better tool that provided genetic variability among okra genotypes.

According to Chahal et al. [32], characters with the largest absolute values closer to unity in the first principal component influence the clustering more than those with lower absolute values closer to zero. Therefore, in the present study, the differentiation of the genotypes into different clusters was because of a cumulative effect of a number of traits rather than the large contribution of a few traits. Days to first flowering, stem diameter, leaf length, number of seeds per pod, seed weight per pod, days to maturity, leaf width, and days to 50% flowering contributed much to PCA1 in the range between 0.836 to 0.917 and average green fruit weight, weight of green fruit per plant, number of green fruits per plant and fruit yield per hectare with PCA2 contribution ranged from 0.885 to 0.937 (Fig 1; "Table 7"). The number of dry pods per plant (0.803) and internode length (0.70) much contributed to PCA3 and PCA4, respectively (Fig 2; "Table 7"). This indicated that these traits had many contributions to the variability of okra genotypes that could be used to evaluate genotypes in the future.

Biplot analysis was carried out based on the first two PCAs. The genotypes and quantitative traits were shown on a biplot to visualize their associations and differences. The first and second PCAs biplot explained 55.56% of the total variability among the genotypes displaying days to first flowering, days to 50% flowering, days to maturity, number of internodes, internode length, stem diameter and green fruit length being considered as the most discriminating traits. The genotypes that were positioned on the left top quadrant were associated and characterized by the tallest green fruit width (GFW), highest weight of green fruit (WGFPP), average green fruit weight (AGF) and green fruit yield (FYD). The genotypes demarcated on the right top quadrant were associated with the highest number of green fruits per plant (NGFPP), longest leaf length (LL), widest leaf width (LW), widest stem diameter (SD), shortest internode length (IL), early in 50% emergence (DE), lowest number of dry pods per plant (NDPPP), the lowest seed yield per plant (SYPP), the highest number of branches per plant (NBPP), weight of dry pod (WDPP), average weight of dry pod (ADPW), number of seed per pod (NDSPP), and seed weight per pod (SWPP). The genotypes were positioned on the right bottom quadrant characterized by late days to 1st (FFL) and 50% flowering (HFL) and days to maturity (DM), longest green fruit length (GFL), lowest seed moisture content (SMC) and hundred seed weight (HSW). The genotypes positioned on the left bottom quadrant characterized by the shortest plant height (PH) and the highest number of internodes (NI). The genotypes concentrated around the origin had similar genetic characteristics, while the genotypes that were found far from the origin are considered as unrelated genotypes [33]. Therefore, the selection of these genotypes as potential parents would result in successful hybridization to develop heterotic groups in the okra breeding program (Fig 1).

PCA3 and PCA4 biplots explained 19% of the total variability among the genotypes, displaying that green fruit width, average green fruit weight, and plant height, seed weight per pod, and seed yield per plant were considered as the most discriminating traits. The genotypes that were positioned on the left top quadrant were closely associated and characterized by late day to 50% emergence (DE) and longest internode length (IL). The genotypes demarcated on the right top quadrant were associated with the longest plant height (PH), highest hundred seed weight (HSW) and seed moisture content (SMC), lowest fruit yield (FYD), shortest leaf length (LL), narrowest leaf width (LW), narrowest green fruits width (GFW), lowest number of green fruits per plant (NGFPP), lowest seed weight per pod (SWPP) and lowest number of

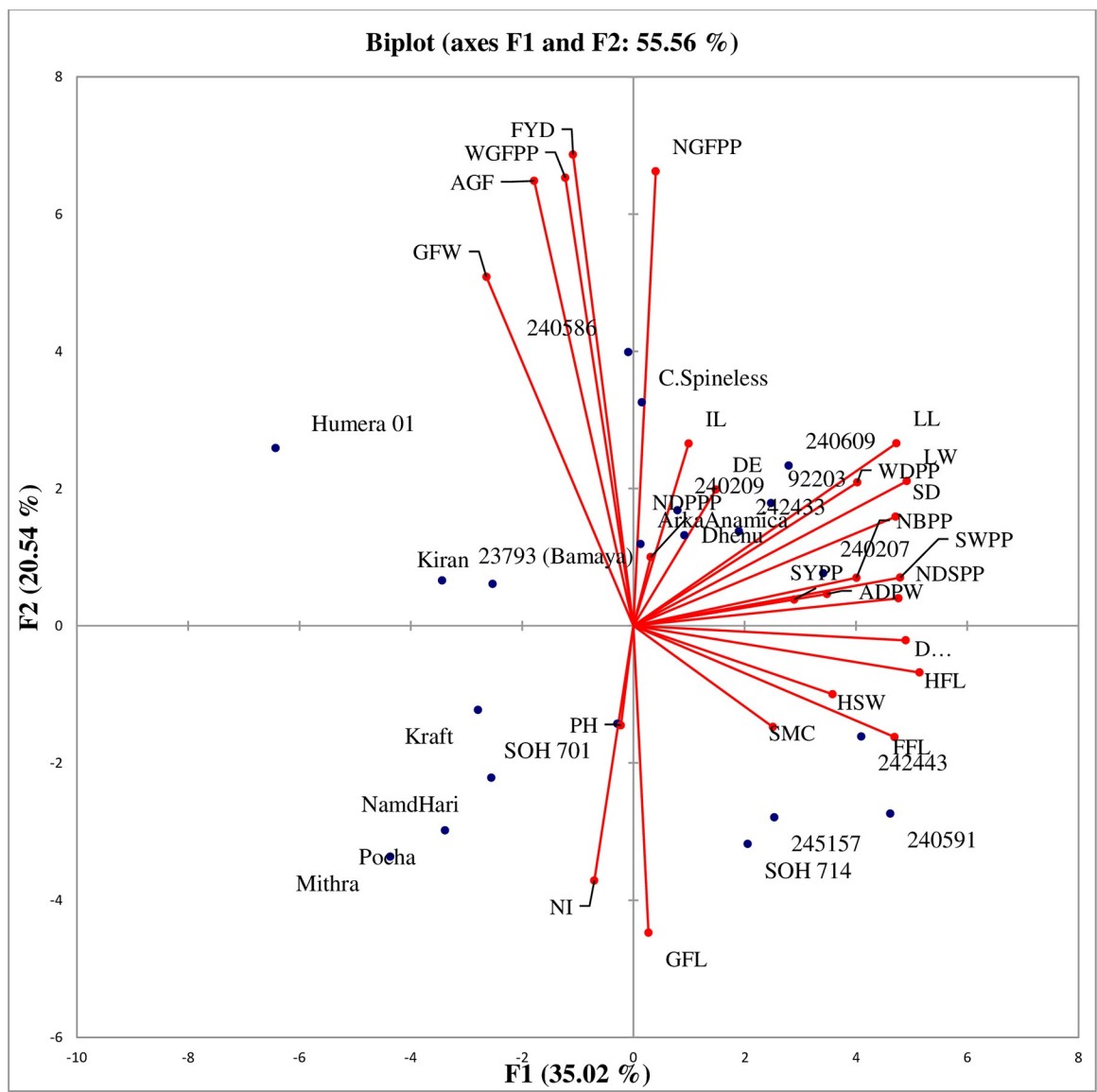

**Fig 1. Biplot PCA1 and PCA 2 of 25 quantitative traits of 21 okra genotypes evaluated at Drie Dawa in 2020.**

seed per pod (NDSPP). The genotypes positioned on the left bottom quadrant characterized by early attainable days to 1st (FFL) and 50% flowering (HFL) and days to maturity (DM) and lowest average dry pod weight (ADPW). The genotypes positioned on the right bottom quadrant characterized by the highest number of dry pods per plant (NDPP), weight of dry pod (WDPP), highest seed yield per plant (SYPP, highest number of branches per plant (NBPP), smallest weight of green fruit (WGFPP) and average green fruit weight (AGFW) Fig 2. These PCA3 and PCA4 biplots provided important information regarding the similarities as well as the pattern of differences among the okra genotypes and of the interrelationships between traits. The okra genotypes scattered in all four quadrants on the axis indicated there was a wide genetic variability for the traits studied. Genotypes that overlapped and closer to each other in the principal component axes had similar genetic makeup. However, genotypes which are far from each other could be considered as genetically distinct [33].

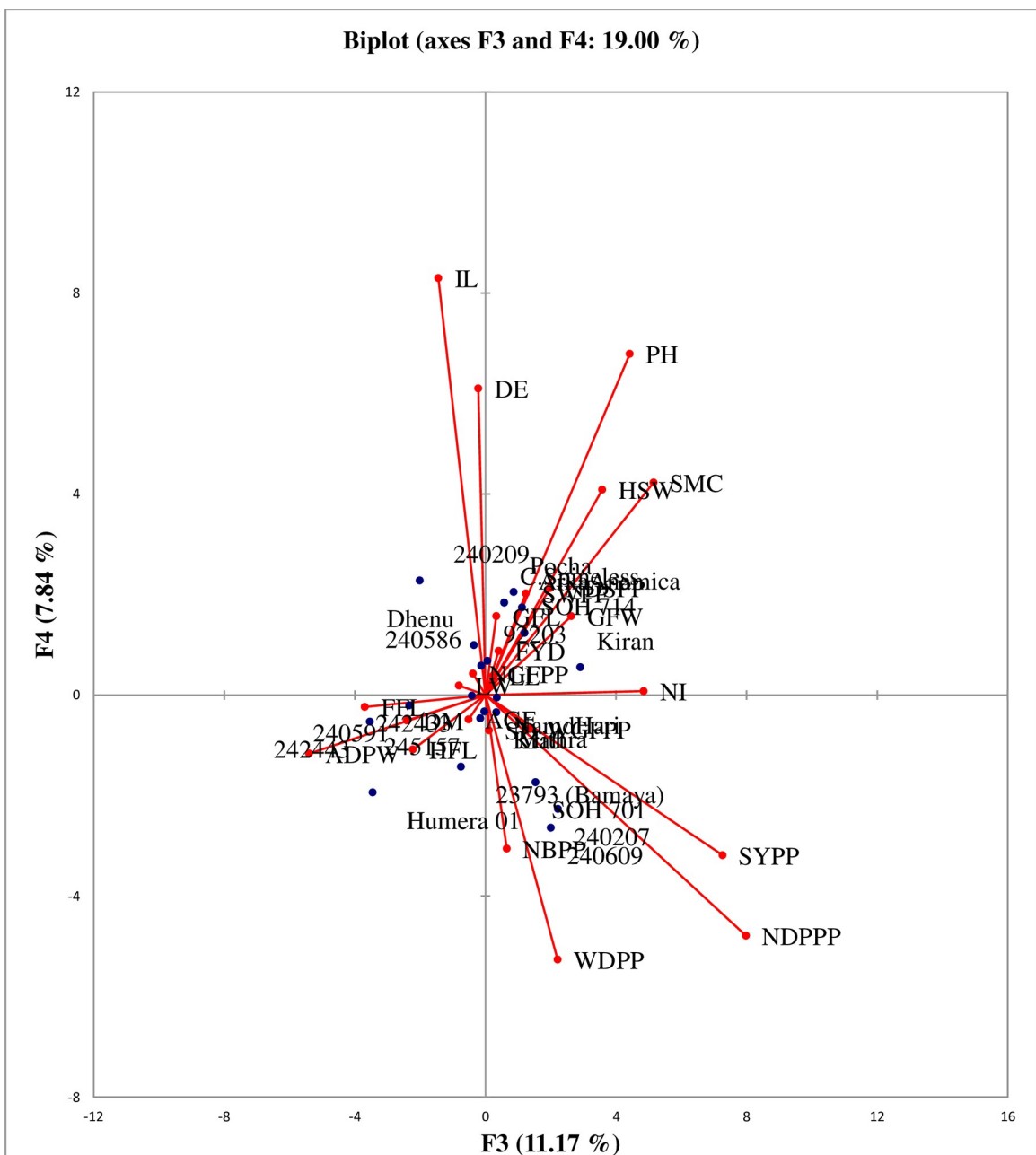

**Fig 2. Biplot PCA3 and PCA 4 of 25 quantitative traits of 21 okra genotypes evaluated at Dire Dawa in 2020.**

## Euclidean distance and cluster analysis

**Euclidean distances among okra genotypes.** The Euclidean distances of all possible 210 pairs of 21 okra genotypes were estimated from 25 quantitative traits and the results are presented in "Table 8". The Euclidean distances of the genotypes ranged from 2.33 to 12.56 with a mean (6.83), standard deviation (1.8), and coefficient of variation (26.46%). The highest Euclidean distance was computed between the genotypes Humera 01 (northern) and 240591 (south-western Ethiopia) 12.56 followed by Humera 01 (northern) and 242443 (western

Ethiopia) 11.82, While, the lowest genetic distance was estimated between Kraft and Namd-Hari (India) 2.33 followed by between 92203 (western Ethiopia) and Dhenu (India) 3.26. Generally, 87 (41.43%) pairs of genotypes had Euclidean distances of 5.03 to 6.83(overall mean of genotypes minus standard deviation), 56 (26.66%) pairs of genotypes had Euclidean distances 6.84 to 8.63 (between mean and overall mean of genotypes plus standard deviation), while 38 (18.1%) pair of genotypes had Euclidean distances >8.63 and 29(13.81%) pair of genotypes had <5.03 Euclidean distances. Generally, the genetic distances measured among the okra genotypes collected from Ethiopia had higher Euclidean distances than the introduced ones. This showed that there is a higher chance of improving okra growth, green fruit yield, seed yield, and related traits through selection and/or hybridization of okra genotypes from different okra growing regions of Ethiopia.

According to YIMAM [16], reported the genetic distance for 24 okra genotypes ranged from 1.96 to 11.36 with a mean, standard deviation, and coefficient of variation of 5.85, 1.97 and 33.75%, respectively. She also observed that genotypes collected from different regions of Ethiopia had higher Euclidean distances than the introduced genotypes. Estimated genetic distances of 25 okra genotypes in which the genetic distance ranged from 3.1 to 12.6 with a mean of 7.2, standard deviation 2, and coefficient of variation of 27.85. They also reported the highest Euclidean distances were observed between genotypes collected from Ethiopia and introduced, while the lowest genetic distance was estimated between introduced genotypes [15]. Genotypes collected from different parts of Ethiopia exhibited wider genetic distances than introduced genotypes [28].

The mean genetic distance of each okra genotype to the other 20 genotypes was calculated to have information about the closest and most distant genotypes "Table 9". Depending on the mean Euclidean distance, Dhenu (India) 5.63, followed by SOH 701 (registered commercial varieties) 5.96, and 92203 (western Ethiopia) 5.97 were found to be the closest to other genotypes, while genotypes collected from south-western, western, and northern Ethiopia were the most distant genotypes to others, 240591 (8.02), 242443 (8.09) and Humera 01 (9.53), respectively. Generally, eight genotypes (38.09%) collected from south-western (240586, 240609, and 240591), western (242443), north-western (240207), northern Ethiopia (Humera 01), and two other genotypes introduced from India (Pocha and Mithra) had mean genetic distances of >6.83 (overall mean genetic distances of genotypes). However, only one genotype collected from northern Ethiopia (Humera 01) had the significantly highest mean genetic distance of >8.63 (overall mean genetic distance of genotype + standard deviation).

The result suggested the presence of a considerable number of distant okra genotypes with others that could be used in a crossing program [16]. Genetic variations are an important feature to achieve the diversified goals of plant breeding, including higher yield, resistance to diseases, quality of the yield, and wider adaptations [34].

**Clustering of genotypes.** The Euclidean distance matrix of 210 pairs of genotypes estimated from 25 quantitative traits was used to construct dendrograms based on the Unweighted Pair-group method with Arithmetic Means (UPGMA). Accordingly, the 21 okra genotypes were grouped into four distinct clusters. Cluster I consisted of 12 genotypes (57.14%) collected from south western (240586 and 240609), western (92203 and 242433), and north western Ethiopia (240209, 240207 and 245157), two registered commercial varieties (SOH 714 and SOH 701), two genotypes introduced from India (Dhenu and ArkaAnamica), and genotype introduced from the USA (Clemson Spineless), while cluster II consisted of two genotypes collected from south western (240591) and western Ethiopia (242443) 9.52%. Cluster III consisted of only one genotype (Humera 01) collected from northern Ethiopia (4.76%) and cluster IV contained six genotypes, one genotype collected from northern Ethiopia

**Table 8. Euclidean distances of 21 okra genotypes based on 25 quantitative traits evaluated at Dire Dawa in 2020.**

| | 240586 | 240609 | 240591 | 242443 | 92203 | 242433 | 240209 | 240207 | 245157 | Humera | Bamaya |
|---|---|---|---|---|---|---|---|---|---|---|---|
| 240586 | | 6.820 | 9.230 | 9.077 | 5.718 | 6.444 | 5.247 | 6.889 | 8.079 | 8.884 | 6.972 |
| 240609 | 6.820 | | 8.355 | 8.064 | 5.069 | 5.167 | 7.286 | 4.846 | 6.657 | 11.021 | 8.106 |
| 240591 | 9.230 | 8.355 | | 5.830 | 6.413 | 6.991 | 6.880 | 6.663 | 4.652 | 12.567 | 9.483 |
| 242443 | 9.077 | 8.064 | 5.830 | | 6.519 | 6.737 | 7.276 | 7.702 | 5.355 | 11.823 | 9.182 |
| 92203 | 5.718 | 5.069 | 6.413 | 6.519 | | 3.673 | 4.386 | 5.142 | 5.349 | 10.115 | 7.080 |
| 242433 | 6.444 | 5.167 | 6.991 | 6.737 | 3.673 | | 4.405 | 5.692 | 5.060 | 9.714 | 6.417 |
| 240209 | 5.247 | 7.286 | 6.880 | 7.276 | 4.386 | 4.405 | | 7.127 | 6.267 | 8.803 | 6.722 |
| 240207 | 6.889 | 4.846 | 6.663 | 7.702 | 5.142 | 5.692 | 7.127 | | 5.475 | 11.632 | 8.024 |
| 245157 | 8.079 | 6.657 | 4.652 | 5.355 | 5.349 | 5.060 | 6.267 | 5.475 | | 11.098 | 7.658 |
| Humera 01 | 8.884 | 11.021 | 12.567 | 11.823 | 10.115 | 9.714 | 8.803 | 11.632 | 11.098 | | 5.950 |
| 23793 | 6.972 | 8.106 | 9.483 | 9.182 | 7.080 | 6.417 | 6.722 | 8.024 | 7.658 | 5.950 | |
| SOH 714 | 8.794 | 7.526 | 6.043 | 6.588 | 6.016 | 6.148 | 6.797 | 6.465 | 3.789 | 11.547 | 8.189 |
| SOH 701 | 7.268 | 6.226 | 7.262 | 7.987 | 5.582 | 5.343 | 6.603 | 5.384 | 4.412 | 9.138 | 5.666 |
| Dhenu | 4.831 | 5.698 | 7.135 | 6.720 | 3.262 | 3.666 | 4.118 | 6.049 | 5.097 | 9.269 | 6.176 |
| Kiran | 6.832 | 7.503 | 10.072 | 9.733 | 6.618 | 7.037 | 7.147 | 7.160 | 7.450 | 8.252 | 5.334 |
| Kraft | 7.180 | 7.670 | 8.669 | 8.670 | 6.463 | 6.584 | 6.672 | 7.864 | 6.023 | 7.253 | 4.986 |
| Pocha | 8.886 | 9.534 | 9.391 | 9.800 | 8.256 | 7.764 | 7.164 | 9.386 | 7.013 | 8.905 | 5.975 |
| Mithra | 9.090 | 10.239 | 9.919 | 10.177 | 9.135 | 8.640 | 8.431 | 9.650 | 7.465 | 8.553 | 5.792 |
| A.Anamica | 5.505 | 6.703 | 7.759 | 8.040 | 4.089 | 5.620 | 4.926 | 6.069 | 6.193 | 9.447 | 6.412 |
| NamdHari | 7.244 | 8.075 | 8.332 | 8.357 | 6.839 | 6.948 | 6.768 | 7.920 | 5.539 | 7.942 | 5.210 |
| C.Spineless | 4.962 | 6.249 | 8.896 | 8.315 | 3.799 | 4.132 | 4.443 | 6.696 | 7.071 | 8.693 | 6.243 |

| | SOH 714 | SOH 701 | Dhenu | Kiran | Kraft | Pocha | Mithra | A. Anamica | NamdHari | C.Spineless |
|---|---|---|---|---|---|---|---|---|---|---|
| 240586 | 8.794 | 7.268 | 4.831 | 6.832 | 7.180 | 8.886 | 9.090 | 5.505 | 7.244 | 4.962 |
| 240609 | 7.526 | 6.226 | 5.698 | 7.503 | 7.670 | 9.534 | 10.239 | 6.703 | 8.075 | 6.249 |
| 240591 | 6.043 | 7.262 | 7.135 | 10.072 | 8.669 | 9.391 | 9.919 | 7.759 | 8.332 | 8.896 |
| 242443 | 6.588 | 7.987 | 6.720 | 9.733 | 8.670 | 9.800 | 10.177 | 8.040 | 8.357 | 8.315 |
| 92203 | 6.016 | 5.582 | 3.262 | 6.618 | 6.463 | 8.256 | 9.135 | 4.089 | 6.839 | 3.799 |
| 242433 | 6.148 | 5.343 | 3.666 | 7.037 | 6.584 | 7.764 | 8.640 | 5.620 | 6.948 | 4.132 |
| 240209 | 6.797 | 6.603 | 4.118 | 7.147 | 6.672 | 7.164 | 8.431 | 4.926 | 6.768 | 4.443 |
| 240207 | 6.465 | 5.384 | 6.049 | 7.160 | 7.864 | 9.386 | 9.650 | 6.069 | 7.920 | 6.696 |
| 245157 | 3.789 | 4.412 | 5.097 | 7.450 | 6.023 | 7.013 | 7.465 | 6.193 | 5.539 | 7.071 |
| Humera 01 | 11.547 | 9.138 | 9.269 | 8.252 | 7.253 | 8.905 | 8.553 | 9.447 | 7.942 | 8.693 |
| Bamaya | 8.189 | 5.666 | 6.176 | 5.334 | 4.986 | 5.975 | 5.792 | 6.412 | 5.210 | 6.243 |
| SOH 714 | | 5.283 | 6.374 | 6.486 | 6.544 | 6.041 | 8.302 | 6.544 | 5.801 | 6.986 |
| SOH 701 | 5.283 | | 5.318 | 5.441 | 4.321 | 6.183 | 6.033 | 5.457 | 4.025 | 6.406 |
| Dhenu | 6.374 | 5.318 | | 6.450 | 5.182 | 7.210 | 7.407 | 3.632 | 5.416 | 3.669 |
| Kiran | 6.486 | 5.441 | 6.450 | | 5.107 | 5.382 | 7.064 | 5.095 | 5.244 | 5.187 |
| Kraft | 6.544 | 4.321 | 5.182 | 5.107 | | 4.890 | 4.302 | 5.238 | 2.331 | 6.238 |
| Pocha | 6.041 | 6.183 | 7.210 | 5.382 | 4.890 | | 4.932 | 6.930 | 4.330 | 7.566 |
| Mithra | 8.302 | 6.033 | 7.407 | 7.064 | 4.302 | 4.932 | | 7.349 | 3.965 | 8.997 |
| ArkaAnamica | 6.544 | 5.457 | 3.632 | 5.095 | 5.238 | 6.930 | 7.349 | | 5.606 | 4.054 |
| NamdHari | 5.801 | 4.025 | 5.416 | 5.244 | 2.331 | 4.330 | 3.965 | 5.606 | | 6.831 |
| C.Spineless | 6.986 | 6.406 | 3.669 | 5.187 | 6.238 | 7.566 | 8.997 | 4.054 | 6.831 | |

**Table 9. Minimum, maximum and mean of Euclidean distance of 21 okra genotypes estimated from 25 quantitative traits as evaluated at Dire Data in 2020.**

| Genotype | Minimum | Maximum | Mean | SD | CV (%) |
|---|---|---|---|---|---|
| 240586 | 4.830 | 9.230 | **7.190** | 1.460 | 20.340 |
| 240609 | 4.840 | 11.021 | **7.340** | 1.650 | 22.530 |
| 240591 | 4.652 | 12.567 | **8.027** | 1.830 | 22.798 |
| 242443 | 5.355 | 11.823 | **8.098** | 1.596 | 19.707 |
| 92203 | 3.262 | 10.115 | 5.976 | 1.792 | 29.991 |
| 242433 | 3.666 | 9.714 | 6.109 | 1.573 | 25.750 |
| 240209 | 4.118 | 8.803 | 6.373 | 1.346 | 21.126 |
| 240207 | 4.846 | 11.632 | **7.092** | 1.689 | 23.816 |
| 245157 | 3.789 | 11.098 | 6.285 | 1.633 | 25.980 |
| Humera 01 | 5.950 | 12.567 | **9.530** | 1.677 | 17.592 |
| 23793 (*Bamaya*) | 4.986 | 9.483 | 6.779 | 1.288 | 18.999 |
| SOH 714 | 3.789 | 11.547 | 6.813 | 1.562 | 22.931 |
| SOH 701 | 4.025 | 9.138 | 5.967 | 1.246 | 20.888 |
| Dhenu | 3.262 | 9.269 | 5.634 | 1.537 | 27.288 |
| Kiran | 5.095 | 10.072 | 6.730 | 1.445 | 21.469 |
| Kraft | 2.331 | 8.670 | 6.109 | 1.591 | 26.036 |
| Pocha | 4.330 | 9.800 | **7.277** | 1.701 | 23.379 |
| Mithra | 3.965 | 10.239 | **7.772** | 1.914 | 24.633 |
| ArkaAnamica | 3.632 | 9.447 | 6.033 | 1.437 | 23.818 |
| NamdHari | 2.331 | 8.357 | 6.136 | 1.671 | 27.234 |
| Clemson Spineless | 3.669 | 8.997 | 6.272 | 1.722 | 27.449 |
| Overall | 2.33 | 12.56 | 6.83 | 1.80 | 26.46 |

SD = standard deviation and CV = coefficient of variation in %.

(*Bamaya Humera)* and the other five introduced from India (Kiran, Kraft, Pocha, Mithra, and NamdHari) 28.57% (Fig 3, "Table 10").

Mohammed et al. [15] studied 25 okra genotypes, of which 11 and 14 were obtained from other countries and three geographic regions of Ethiopia, respectively. The genotypes were grouped into seven major clusters. They also indicated that genotypes from the same countries tend to be grouped in the same clusters. Yimam [16] also studied 24 okra genotypes, of which 10 and 14 were obtained from other countries and from Ethiopia, respectively. The genotypes were grouped into seven major clusters. Demelie et al. [28] studied 25 okra genotypes and identified ten major clusters, while [4] reported five divergent groups of 25 okra collections from two regions (Gambella and Asossa) of Ethiopia. Binalfew et al. [14] were able to group 50 okra collections into four major clusters, which were obtained from four major production regions of Ethiopia.

**Cluster mean analysis.** The mean values of the four clusters for 25 traits (phenology, growth, green fruits, seed yield and related traits) are presented in "Table 11". The unique features of Clusters III and IV are characterized by the lowest mean values of crop phenology (early on attain, 50% emergence, 1st and 50% flowering, and days to maturity). Clusters I and IV were distinguished by having higher mean values of plant height, leaf length, number of dry pods, and seed yield per plant. Clusters I and II are also characterized by the highest mean values of all growth traits except plant height and leaf length. Clusters I and III were distinguished by having higher mean values of all green fruit related traits (number of green fruits per plant, weight of green fruit per plant, green fruit width, average green fruit weight, and fruit yield per hectare) except green fruit length. Clusters II and IV were characterized by having higher

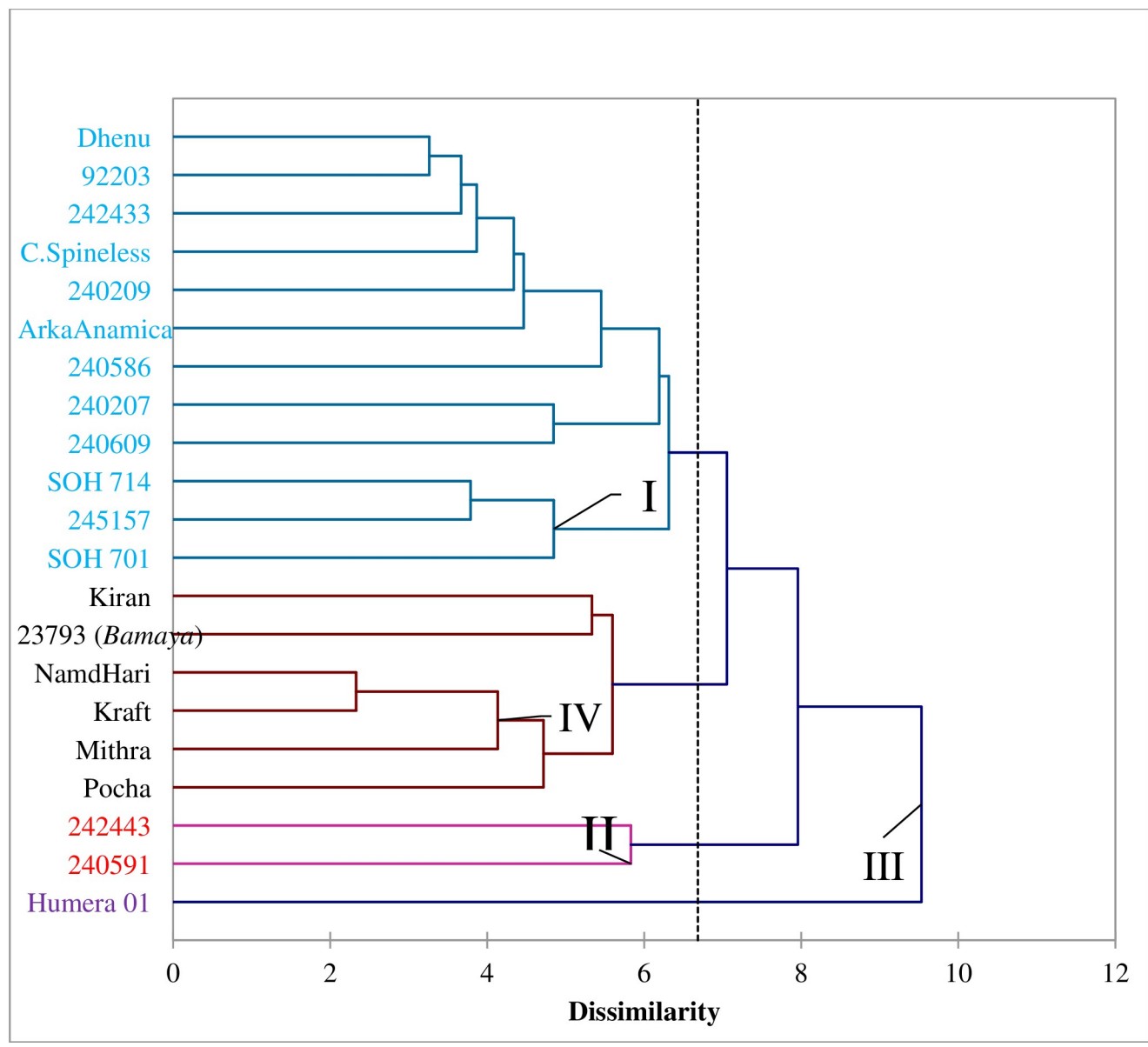

**Fig 3. Dendrogram illustrating dissimilarity of 21 okra genotypes by Unweighted Pair Group Method with Arithmetic Means (UPGMA) clustering method with Euclidean distance matrix based on 25 traits evaluated at Dire Dawa in 2020.**

mean values of green fruit length, while Clusters II and III were characterized by a lower mean value of seed moisture content. These results suggested a higher chance of developing varieties through selection and further evaluation of genotypes from these two clusters (Clusters I and III).

According to the mean analysis of clusters, it is possible to select genotypes from Clusters I and III to obtain genotypes with the highest green fruit yield and other desirable traits. It is also suggested to make crosses between the two cluster members and genotypes from Cluster I to combine desirable traits in hybrids and search for better-performing varieties in subsequent segregating generations. Some of the clusters constructed by okra genotypes obtained from Ethiopia had higher mean values for desirable traits including fruit yield, seeds per pod, and hundred seed weight [4,14,15,28].

**Table 10. Number of genotypes grouped in four clusters, genotypes, and collection region of 21 okra genotypes evaluated at Dire Dawa in 2020.**

| Clusters | Number of Genotype | Genotypes | Geographic Origin |
|---|---|---|---|
| I | 12 | 240586 and 240609 | South western |
| | | 240209 240207 and 245157 | North western and western |
| | | 92203 and 242433 | Western |
| | | Dhenu, ArkaAnamica | India |
| | | Clemson Spineless | USA |
| | | SOH 714 and SOH 701 | Registered commercial verities |
| II | 2 | 240591, and | South western (Akobo) |
| | | 242443 | Western (Menge) |
| III | 1 | Humera 01 | Northern (Tigray) |
| IV | 6 | 23793 (*Bamaya Humera*) Kiran, Kraft, Pocha, Mithra, NamdHari | Northern India |

## Conclusion

Generally, the research results indicated the following conclusion: i) there are wide variations among okra genotypes for fruit yield, seed yield, and yield related traits ii), Okra genotypes collected from Ethiopia had higher performance for most of the traits, including growth, green fruit yield, and seed yield, than introduced genotypes. iii), Okra genotypes collected from

**Table 11. Cluster mean values for 25 quantitative traits of 21 okra genotypes evaluated at Dire Dawa in 2020.**

| Traits | Cluster | | | |
|---|---|---|---|---|
| | I | II | III | IV |
| Days to 50% emergence | 11.44 | 10.67 | 9.33 | 10.33 |
| Days to first flowering | 54.31 | 73.70 | 44.70 | 45.93 |
| Days to 50% of flowering | 68.54 | 83.70 | 51.70 | 53.45 |
| Days to maturity | 84.76 | 101.70 | 64.30 | 64.31 |
| Plant height (cm) | 155.70 | 140.15 | 138.17 | 159.52 |
| Stem diameter (cm) | 3.61 | 3.70 | 2.76 | 2.87 |
| Number of branches per plant | 4.66 | 4.50 | 2.20 | 2.87 |
| Number of internodes per plant | 20.62 | 21.25 | 15.20 | 16.38 |
| Internode's length (cm) | 25.35 | 28.18 | 19.16 | 20.01 |
| Leaf length (cm) | 31.00 | 29.29 | 28.06 | 34.02 |
| Leaf width (cm) | 5.05 | 5.09 | 4.93 | 4.67 |
| Number of green fruits per plant | 16.57 | 12.05 | 17.24 | 12.86 |
| Weight of green fruit per plant (g | 312.92 | 146.99 | 458.30 | 262.31 |
| Green fruit length (cm) | 13.78 | 17.03 | 11.42 | 16.44 |
| Green fruit width (cm) | 2.17 | 1.58 | 2.51 | 2.13 |
| Average green fruit weight (g) | 36.76 | 22.98 | 53.26 | 33.43 |
| Fruit yield per hectare (t ha-1) | 6.87 | 3.06 | 9.54 | 5.46 |
| Number of dry pods per plant | 16.50 | 11.16 | 11.33 | 16.40 |
| Weight of dry pods per plant (g) | 780.15 | 897.85 | 524.00 | 490.35 |
| Average dry pod weight (g) | 47.70 | 91.33 | 46.33 | 30.08 |
| Number of seeds per pod | 91.33 | 90.30 | 31.67 | 58.87 |
| Hundred seed weight (g) | 6.36 | 6.76 | 2.33 | 5.84 |
| Seed moisture content (%) | 10.70 | 9.69 | 3.25 | 10.18 |
| Seed weight per pod (mg) | 5857.41 | 6521.95 | 1933.30 | 3728.06 |
| Seed yield per plant (g) | 53.57 | 35.510 | 10.66 | 36.80 |

Ethiopia were more divergent with high genetic distances than the introduced and two registered commercial varieties. Therefore, the indigenous okra genotypes were superior in most of the traits and more important than those introduced. These major results suggested a higher chance of developing okra varieties through selection within the okra genotypes collected from Ethiopia and other countries. However, the evaluation was conducted under non-stress conditions with sufficient irrigation water supply at one location. Therefore, it is recommended to conduct the evaluation of genotypes under controlled and random stresses over location and season.

## Acknowledgments

The authors would like to express gratitude to the administrators of the Tony Farm Research Substation of Harmaya University and field data collectors for their cooperation.

## Author Contributions

**Conceptualization:** Wubadis Kenaw.

**Data curation:** Wubadis Kenaw.

**Formal analysis:** Wubadis Kenaw.

**Investigation:** Wubadis Kenaw.

**Methodology:** Wubadis Kenaw.

**Project administration:** Wubadis Kenaw.

**Supervision:** Wubadis Kenaw, Wassu Mohammed, Kebede Woldetsadik.

**Validation:** Wubadis Kenaw, Wassu Mohammed, Kebede Woldetsadik.

**Visualization:** Wassu Mohammed, Kebede Woldetsadik.

**Writing – original draft:** Wubadis Kenaw.

**Writing – review & editing:** Wubadis Kenaw.

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
