## [Decision Letter · Decision Letter 0]

17 May 2023

PONE-D-23-10456Morpho-Agronomic Variability of Okra [Abelmoschus Esculentus (L.) Moench] Genotypes of Diverse Origin in Dire Dawa, Eastern EthiopiaPLOS ONE

Dear Dr. kenaw,

Thank you for submitting your manuscript to PLOS ONE. After careful consideration, we feel that it has merit but does not fully meet PLOS ONE’s publication criteria as it currently stands. Therefore, we invite you to submit a revised version of the manuscript that addresses the points raised during the review process.

We look forward to receiving your revised manuscript.

Kind regards,

Mehdi Rahimi, Ph.D.

Academic Editor

PLOS ONE

Journal Requirements:

Reviewers' comments:

Reviewer's Responses to Questions

**Comments to the Author**

1. Is the manuscript technically sound, and do the data support the conclusions?

Reviewer #1: Partly

Reviewer #2: No

Reviewer #3: Yes

2. Has the statistical analysis been performed appropriately and rigorously? 

Reviewer #1: Yes

Reviewer #2: Yes

Reviewer #3: Yes

3. Have the authors made all data underlying the findings in their manuscript fully available?

Reviewer #1: No

Reviewer #2: Yes

Reviewer #3: Yes

4. Is the manuscript presented in an intelligible fashion and written in standard English?

Reviewer #1: No

Reviewer #2: No

Reviewer #3: Yes

5. Review Comments to the Author

Reviewer #1: The research is a very good one but not concisely and clearly presented The authors addressed too many things at a time. The manuscript is too lengthy and should be reduced and made clear as much as possible.

Reviewer #2: I have read the paper written by Kenaw et al. and they investigated the agro-morphological and genetic variability of diverse okra genotypes based on growth and yield attributes. Although the methods and results of the study is appropriate for a single year/season experiment, but I cannot recommend this article for publication in the Journal (PlosOne) in the present format due to validity of the data and novelty of the work are below the standard level. My major concern is, yield and productivity data (for concluding according to the objectives of this research) should be collected at least for two years/seasons and the experiment must be undertaken in multiple locations. And thereafter, advance statistical analysis (mixed model and multivariate analyses) is necessary to extract data precisely and draw conclusions properly. The English language of the article need to be improved considerably. I, therefore, recommend repeating the experiments as per suggestions mentioned above.

Reviewer #3: The authors need to include all suggestions given in attached manuscript PDF. Add more latest references in Discussion section. Correct the tables according given suggestions of the reviewers.

The title should be corrected as "Morpho-Agronomic Variability of Okra [Abelmoschus Esculentus (L.) Moench] Genotypes in Dire Dawa, Eastern Ethiopia"

Add SEM or SD values along with each mean, Add CV(%) values in table 2,3, 4

What does the alphabets attached to mean values indicate for what information

6. PLOS authors have the option to publish the peer review history of their article (what does this mean?). If published, this will include your full peer review and any attached files.

Reviewer #1: No

Reviewer #2: No

Reviewer #3: **Yes: **Jiban Shrestha

---

## [Author Response · Author response to Decision Letter 0]

25 May 2023

Author’s response to editor and reviewers

For Academic editor

Comment 1: when submitting your revised manuscript include A rebuttal letter, A marked-up copy of your manuscript that highlights changes and an unmarked version of your revised paper without tracked changes

Response 1: thank you for your recommendation. Now we uploaded all file accordingly.

Comment 2: Please ensure that your manuscript meets PLOS ONE's style requirements, including those for file naming.

Response 2: thank you for your comment. Now the revised version is fulfilled style requirements.

Comment 3: PLOS requires an ORCID iD for the corresponding author in Editorial Manager on papers submitted after December 6th, 2016. Please ensure that you have an ORCID iD and that it is validated in Editorial Manager

Response 3: thank you for your recommendation. Now the ORCID iD created for the corresponding author

Comment 4: Response 4: No change at all.

 For reviewer 1

Comment 1: The research is a very good one but not concisely and clearly presented. The authors addressed too many things at a time. The manuscript is too lengthy and should be reduced and made clear as much as possible.

Response 1: Thank you for your constructive comment. We tried to reduce the manuscript and make more clear.

 For reviewer 2

Comment 1: My major concern is, yield and productivity data (for concluding according to the objectives of this research) should be collected at least for two years/seasons and the experiment must be undertaken in multiple locations. And thereafter, advance statistical analysis (mixed model and multivariate analyses) is necessary to extract data precisely and draw conclusions properly. 

Response 1: Thank you for your comment. Initially, we proposed that the experiment be conducted at two sites. But, due to the lockdown of the COVID-19 pandemic, the experiment was conducted in only one location and we tried a lot to clearly present the available data with suitable statistical analysis.

Comment 2: The English language of the article need to be improved considerably. I, therefore, recommend repeating the experiments as per suggestions mentioned above.

Response 2: Thank you for your recommendation. The revised version of main document edited by English language expert person.

 For reviewer 3

Comment 1: The authors need to include all suggestions given in attached manuscript PDF. Add more latest references in Discussion section. Correct the tables according given suggestions of the reviewers.

Response 1: thank you for your constructive comments. We incorporated all comments suggested in the above. 

Comment 2: The title should be corrected as "Morpho-Agronomic Variability of Okra [Abelmoschus Esculentus (L.) Moench] Genotypes in Dire Dawa, Eastern Ethiopia"

Response 2: Thank you for your constructive comments: we corrected as commented 

Comment 3: What does the alphabets attached to mean values indicate for what information

Response 3: Thank you for your comment. We indicated the information for alphabets for what was attached with the mean values in the main document at table 2, 3 and 4 footnotes’

---

## [Decision Letter · Decision Letter 1]

29 Jun 2023

Morpho-agronomic variability of okra [Abelmoschus esculentus (L.) Moench] genotypes in Dire Dawa, eastern Ethiopia

PONE-D-23-10456R1

Dear Dr. kenaw,

We’re pleased to inform you that your manuscript has been judged scientifically suitable for publication and will be formally accepted for publication once it meets all outstanding technical requirements.

Kind regards,

Mehdi Rahimi, Ph.D.

Academic Editor

PLOS ONE

Additional Editor Comments (optional):

Reviewers' comments:

Reviewer's Responses to Questions

**Comments to the Author**

1. If the authors have adequately addressed your comments raised in a previous round of review and you feel that this manuscript is now acceptable for publication, you may indicate that here to bypass the “Comments to the Author” section, enter your conflict of interest statement in the “Confidential to Editor” section, and submit your "Accept" recommendation.

Reviewer #3: All comments have been addressed

2. Is the manuscript technically sound, and do the data support the conclusions?

Reviewer #3: Yes

3. Has the statistical analysis been performed appropriately and rigorously? 

Reviewer #3: Yes

4. Have the authors made all data underlying the findings in their manuscript fully available?

Reviewer #3: Yes

5. Is the manuscript presented in an intelligible fashion and written in standard English?

Reviewer #3: Yes

6. Review Comments to the Author

Reviewer #3: The revised version of manuscript "Morpho-agronomic variability of okra [Abelmoschus esculentus (L.) Moench] genotypes

in Dire Dawa, eastern Ethiopia" looks good to accept.

7. PLOS authors have the option to publish the peer review history of their article (what does this mean?). If published, this will include your full peer review and any attached files.

Reviewer #3: No

---

## [Editor Report · Acceptance letter]

7 Jul 2023

PONE-D-23-10456R1 

Morpho-agronomic variability of okra [*Abelmoschus esculentus* (L.) Moench] genotypes in Dire Dawa, eastern Ethiopia 

Dear Dr. Kenaw:

I'm pleased to inform you that your manuscript has been deemed suitable for publication in PLOS ONE. Congratulations! Your manuscript is now with our production department. 

Kind regards, 

on behalf of

Associate Prof. Mehdi Rahimi 

Academic Editor

PLOS ONE